# On the dynamic reconfigurable implementations of MISTY1 and KASUMI block ciphers

Huang Jiexian[1], Yasir Khizar[2], Zain Anwar Ali[1,3]*, Raza Hasan[4], Muhammad Salman Pathan[5]

1 School of Physics and Electronic Engineering, JiaYing University, Meizhou, Guangdong, China, 2 School of Electronic Information, Nanjing University of Aeronautics & Astronautics, Nanjing, Jiangsu, China, 3 Electronic Engineering Department, Sir Syed University of Engineering & Technology, Karachi, Sindh, Pakistan, 4 Department of Science and Engineering, Solent University, Southampton, United Kingdom, 5 Computer Science Department, Maynooth University, Co. Kildare, Ireland

* zainanwar86@hotmail.com

## Abstract

Novel hardware architectures for dynamic reconfigurable implementation of 64-bit MISTY1 and KASUMI block ciphers are proposed to enhance the performance of cryptographic chips for secure IoT applications. The SRL32 primitive (Reconfigurable Look up Tables—RLUTs) and DPR (Dynamic Partial Reconfiguration) are employed to reconfigure single round MISTY1 / KASUMI algorithms on the run-time. The RLUT based architecture attains dynamic logic functionality without extra hardware resources by internally modifying the LUT contents. The proposed adaptive reconfiguration can be adopted as a productive countermeasure against malicious attacks with the added advantage of less reconfiguration time (RT). On the other hand, the block architecture reconfigures the core hardware by externally uploading the partial bit stream and has significant advantages in terms of low area implementation and power reduction. Implementation was carried out on FPGA, Xilinx Virtex 7. The results showed remarkable results with very low area of 668 / 514 CLB slices consuming 460 / 354 mW for RLUT and DPR architectures respectively. Moreover, the throughput obtained for RLUT architecture was found as 364 Mbps with very less RT of 445 nsec while DPR architecture achieved speed of 176 Mbps with RT of 1.1 msec. The novel architectures outperform the stand-alone existing hardware designs of MISTY1 and KASUMI implementations by adding the dynamic reconfigurability while at the same achieving high performance in terms of area and throughput. Design details of proposed unified architectures and comprehensive analysis is described.

## Introduction

To meet the IoT security application requirements of wireless networks, cryptographic algorithms have emerged as the key component in SoC designs. To date, various encryption algorithms SNOW, AES, KASUMI, ZUC, MISTY1, etc. have been developed to ensure data confidentiality / integrity. The usage / implementation of these algorithms, however, mainly depends upon the applications and security requirements. In recent times, the dynamic

**Data Availability Statement:** All relevant data are within the paper.

**Funding:** This study was funded by Jiaying University, Minghai Industrial Robot Production

and Education Integrated Base in the form of a grant to HJ [422A0303].

**Competing interests:** The authors have declared that no competing interests exist.

reconfigurable cryptographic computations are proven to be of greater advantages against the hardware Trojans. This study investigates FPGA based reconfigurable hardware implementation of MISTY1 and KASUMI algorithms exploring SRL32 primitive and DPR technology. The proposed work has a broad range of applications including military equipment, sensor networks and wireless communications [1–6].

FPGAs are the widely being adopted platforms for cryptographic circuit implementation due to their prospective features for high-end flexibility, sophisticated on-chip interconnects and run-time reconfigurable computing. In addition, the run-time reconfiguration of FPGAs with RLUT (SRL32 primitive) and DPR has certainly aided towards secure design package. The RLUT reconfigures the hardware core by internally modifying the pre-defined LUT contents. On the other hand, the DPR scheme allows a part of bit stream to be uploaded dynamically at predefined stages / set-timings without arbitration of the running FPGA operation. This allows the FPGAs for hardware implementations in hostile environments [7–10].

MISTY1 / KASUMI algorithms are designed to encrypt 64-bit block of data. MISTY1 is an ISO standardized block UMTS cipher developed by Mitsubishi Electric and is widely being employed in Japanese ATM networks. Followed by MISTY1, KASUMI is a 3gpp core algorithm deployed for confidentiality / integrity of user in 3g Universal Mobile Telecommunications System (UMTS) networks. The two algorithms resemble in terms of Linear / Differential cryptanalysis, architectural characteristics, and the key requirements; nevertheless, they have several contrasting features / design constraints for round operations. The attacks against MISTY1 and 3gpp standardized KASUMI have identified the weaknesses; however, no serious drawbacks / shortcomings have been found against the full 8-rounds MISTY1 and 3gpp standardized KASUMI output feedback (OFB) and Cipher Block Chaining (CBC) modes [11–14].

A detailed investigation has been carried out on the optimization techniques of MISTY1 and KASUMI algorithms, reconfigurable hardware implementations, PUFs implementation, DPR methodology and SRL32 primitive. Regarding MISTY1 and KASUMI implementations, variety of high speed and low area hardware architectures have been proposed till date for compact ASIC applications and sensor networks. A deep insight reveals that the optimization techniques adopted on MISTY1 can be employed to its counterpart KASUMI with essential modifications. Moreover, the common logic functions can be utilized for reconfigurable hardware implementation. In this regard, a highly area-efficient implementation of reconfigurable MISTY1 / KASUMI is proposed in constituting 3481 NAND gates; however, the hardware is based on static reconfiguration. Thus, the entire bit-stream and logic functionality / selection methodology has to be uploaded on the initial-run of FPGA and is, therefore, more prone to attacks. Moreover, the repetitive-loop structure results in a low throughput value of 130.2 Mbps and in-turn the high-power consumption [15–26]. Studies on static reconfigurable cryptographic modules have also been proposed for SNOW / ZUC algorithms and AES / KASUMI / SNOW / ZUC algorithms signifying flexible implementations [14, 27–36].

Compared to static hardware circuits, dynamic reconfigurable implementations appear with the added advantages of security and power reduction. The SRL32 primitive has been reported for applications pertaining to constructive as well as destructive usages. The constructive applications include the dynamic operability of LUTs whereas the destructive application appears in the form of hardware Trojans insertion for the retrieval of secret keys. On the other hand, the DPR methodology has been adopted by designers to employ multiple algorithms. The study is proposed in employed DPR as the core hardware for implementation of PUF-- HELP and AES. The DPR technology employing AES, TDEA, MISTY1, CAMELLIA, SEED and CAST-128 has also been reported in representing power efficient design. In addition, the technique has also been employed to implement the variants of AES i.e., AES– 128 / 192 and 256 bits [27, 28, 30].

The studies reported to date have either employed the low area or high speed techniques for MISTY1/KASUMI implementations. No studies have been found which have identified the similarities of the two 64-bit MISTY1 and KASUMI algorithms and implement them for dynamic reconfigurability. The same has been implemented with while keeping in view the key advantages of SRL32 primitive and DPR architecture. Dynamic reconfigurable MISTY1 / KASUMI architectures have implemented with improved performance in terms of occupied area, power consumption, RT, and throughput [21–24].

The key contributions of paper include:

1. Design and configuration of RLUT based MISTY1 / KASUMI architecture.

   - Implementation of MISTY1 / KASUMI transformation functions (FL, FI, FO, and 32-bit XOR) and S9 / S7 s-boxes by SRL32.

   - Novel methodology for dynamic reconfiguration of RLUT architecture by utilizing the counter data for compact implementation.

2. Implementation of DPR based reconfigurable hardware architecture.

   - Optimization / configuration of static part utilizing the common logic for single round MISTY1 / KASUMI algorithm.

   - Reconfigurable logic designing of MISTY1 / KASUMI logic functions to complement the static logic for low power implementation.

The paper is preceded by brief description to reconfiguration methodologies and covers the review on MISTY1 and KASUMI algorithms followed by the proposed RLUT and DPR based designs respectively. Finally, the results are summarized with conclusion.

## RLUT, DPR and review of MISTY1 / KASUMI algorithms

### RLUT and DPR

The RLUT and DPR methodology for FPGAs is depicted in Figs 1 and 2 respectively.

The SRL32 consists of 5–1 LUT or 4–2 LUTs with reconfiguration data enable signal (En), clock, reconfiguration data input (RDI) and reconfiguration data output (RDO). The RDI is serially shifted to the LUT on clock edge and high En whereas the old value is depleted from LUT through RDO. The RDI of RLUTs may comprise of the output values of existing LUTs

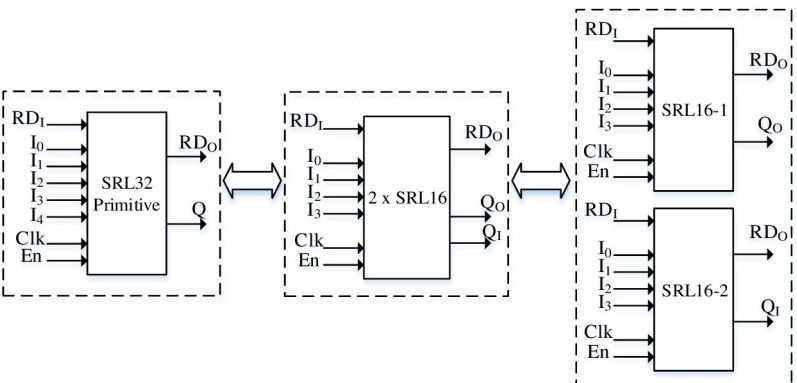

**Fig 1. RLUT (configured as 1 × SRL32 primitive or 2 × SRL16).**

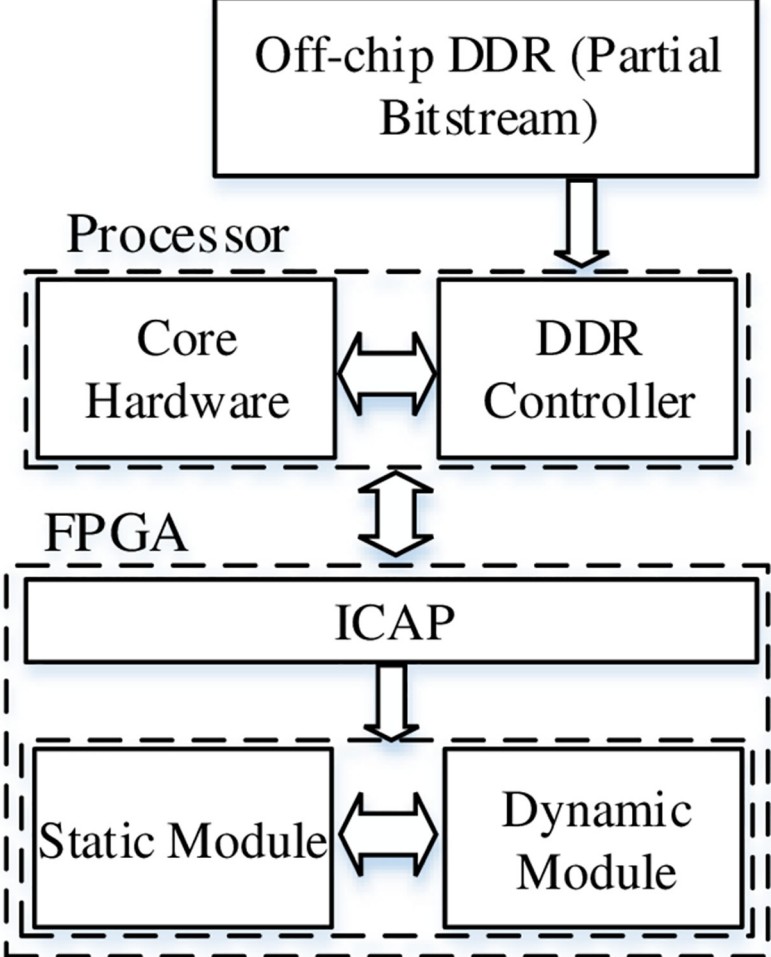

**Fig 2. DPR methodology in FPGAs.**

and thus can be connected in cascaded form [25]. This methodology reconfigures the LUTs internally i.e. no external bit stream is uploaded to modify the hardware. The RT of RLUT architecture depends upon path delay and the number of clock cycles required for re-configuration of LUTs. On the other hand, DPR technology consists of static / dynamic modules with static part representing the logic circuit and is uploaded during the initial run of FPGA. The partial bit stream stored in external DDR (memory) can be dynamically loaded into FPGA core to update the hardware circuit. The DPR supports the uploading baud rate of 400 Mbps and therefore the RT depends upon the size of partial bit stream. The greater the size of partial bit stream, the higher is the RT [26].

### Review of MISTY1 / KASUMI algorithms

A brief review on the similarities and contrasting features of MISTY1 / KASUMI algorithms is presented in Table 1. The algorithms and their transformation functions FL, FO, FI and S9 / S7 s-boxes differ in terms of functionality. In addition, the extended key generation and key

**Table 1. Similarities and differences of MISTY1 and KASUMI.**

| Arch. / Ftn | Similarities | Differences | |
|---|---|---|---|
| | | MISTY1 | KASUMI |
| Algorithm | 64-bit block cipher with 128-bit secret key and 128-bit extended key | MISTY1 comprises additional round consisting of 2 × FL functions | - |
| | Repeated loop characteristic after consecutive 2 rounds. | A 64-bit cipher text consisting of 32-bit LSB and MSB outputs are swapped for MISTY1 | - |
| | Cryptanalysis parameter | Odd Rounds: 2 × FL, FO, 32-bit XOR | Odd Rounds: FL, FO and XOR |
| | | Even Rounds: FO and 32-bit XOR | Even Rounds: FO, FL and 32-bit XOR |
| FL | Combinational logic of 16-bit AND, OR and XOR | - | 2 × 1-bit circular left shift rotation as an additional function |
| FI | 3-rounds feistel-like structure | S9 and S7 s-boxes logic functionality | S9 and S7 s-boxes logic functionality |
| | | - | Additional feistel-round comprising S7 s-box and respective truncated XOR operation |
| FO | 3-rounds feistel-like structure | Additional $KO_{i4}$ XOR operation | - |
| XOR | Logic 32-bit XOR | Inputs / Outputs to XOR | Inputs / Outputs to XOR |
| Ext. Key Generation | - | Generated by FI function | Generated by 16-bit XORs |
| Key Scheduling | - | - | Secret keys are applied after 1, 5, 8 and 13-bits circular left shift operation |
| | | Order of the keys | Order of the keys |

scheduling of MISTY1 is entirely different to that of KASUMI. A detailed specification of MISTY1 / KASUMI algorithms is mentioned.

## RLUT based reconfigurable MISTY1 / KASUMI

The RLUT architecture is depicted in Fig 3 consisting of single round MISTY1 / KASUMI algorithm, EK generation / scheduling and controller. The architecture is modeled with SRL32 primitive and dynamically configures the LUTs based on enable M1 / K and reconfiguration data (RD) signals. The RLUT architecture has been designed to implement MISTY1 / KASUMI logic functions by employing SRL32 such that no additional hardware resources are required. The proposed hardware design is instantiated for MISTY1 algorithm and is dynamically operated for KASUMI.

The input to the architectures includes 64-bit plain text, 128-bit secret key and control signals. The EKs required for 8-rounds MISTY1 / KASUMI algorithm are generated by single round algorithm (by FO function) before the round operations. In order to reconfigure the circuit, the SRL controller dynamically enables the configuration signals using pre-set timings. A 128-bit extended key and secret key is fed through the key scheduling scheme to RLUT based MISTY1 / KASUMI algorithm. A detailed explanation to single round algorithm, its transformation functions and key scheduling scheme is as under.

### A single round MISTY1 / KASUMI algorithm

A single round RLUT based reconfigurable MISTY1 / KASUMI algorithm is presented in Fig 4. The repeated iterations of single round algorithm encipher the plain text and is saved in 2 × 32-bit registers R1 and R2 after 9 and 8 clock cycles for MISTY1 and KASUMI algorithms respectively. The additional clock cycle for the 8-round MISTY1 algorithm is due to the last (i.e. 9th) round consisting of 2 × FL functions. Besides, 4 × clock cycles are required to generate the extended for MISTY1 / KASUMI round operations.

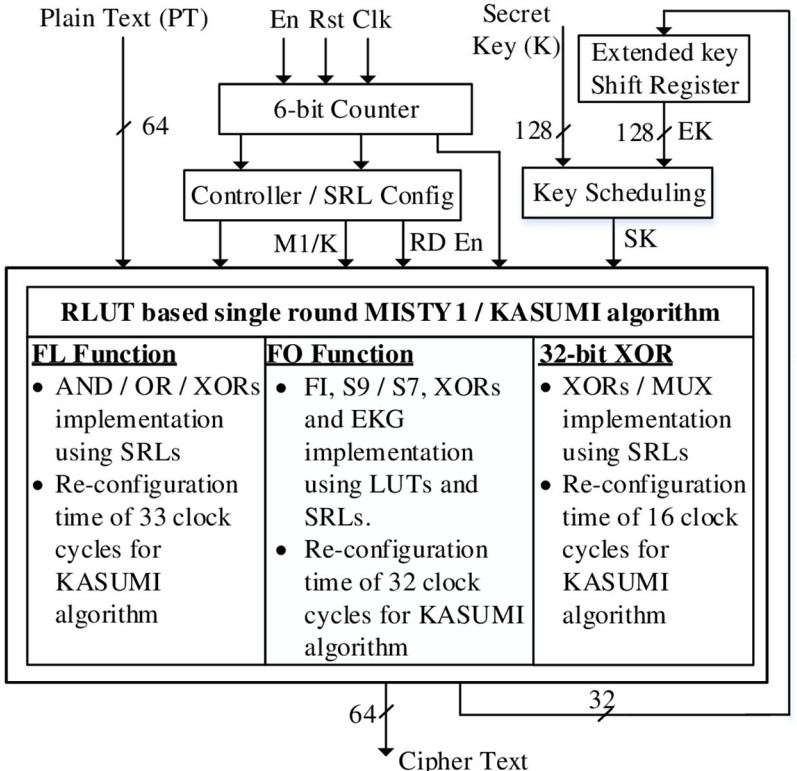

**Fig 3. RLUT architecture.**

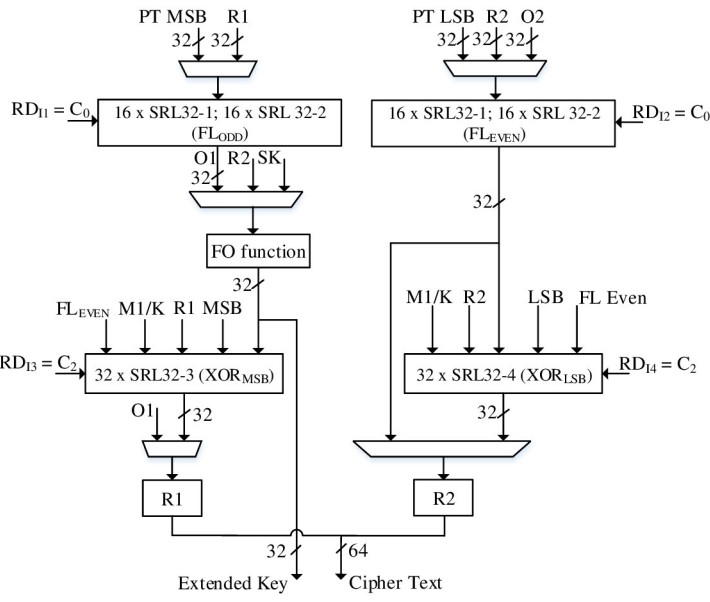

**Fig 4. Single round RLUT based MISTY1 / KASUMI algorithm.**

**Table 2. Combination of input signals for respective output signals.**

| Ftn | SRL32 Primitive | Input Signal | Output Signal |
|---|---|---|---|
| FL | SRL32-1 (LSB) | $\{KL_{L15}, X_{L15}, KL_{L0}, X_{L0}, X_{R0}\}$ | $Y_{R0}$ |
| | | $\{KL_{L(1+1)}, X_{L(1+1)}, KL_{LI}, X_{LI}, X_{RI}\}$ | $Y_{R1}$—$Y_{R15}$ |
| | SRL32-2 (MSB) | $\{KL_{R15}, Y_{R15}, KL_{R0}, Y_{R0}, X_{L0}\}$ | $Y_{L0}$ |
| | | $\{KL_{R(1+1)}, Y_{R(1+1)}, KL_{RI}, Y_{RI}, X_{LI}\}$ | $Y_{L1}$—$Y_{L15}$ |

The architectures characterize SRL32 primitive for FL function and 32-bit XOR implementation. The FL function consists of 16 × SRL32-1 and 16 × SRL32-2 for LSBs and MSBs respectively with initial values representing MISTY1 logic operation. The RLUTs are designed such that the re-configuration of LUTs can be carried out for FL function KASUMI operation without additional logic. This is obtained by deriving the relation for the input bits to formulate the RLUTs for MISTY1 and KASUMI FL function operations and is given in Table 2 as under.

For each output bit, the RLUTs input signals are applied based on the combination given in Tab. 2. The initial values stored in RLUTs comprise of MISTY1 FL function. Thus, in order to configure the circular left shift operation of FL function for KASUMI algorithm, the SRL32 values are updated by RDI1 and RDI2 connected to counter C0 signal. Thus, FL function can be implemented without additional hardware cost using the counter of RLUT architecture. In a similar way, a 32-bit XOR implementation is carried out by employing 2 × SRL32 sets i.e., SRL32-3 and SRL32-4 executing odd and even round XOR operations respectively. The SRL32 for XOR operation is implemented as 2 × SRL16 generating 2 × outputs. The $RD_{I3}$ and $RD_{I4}$ are connected to $C_2$ signal of the counter and update the LUTs by retaining the 16-bit initial values. The use of RLUTs for 32-bit XOR implementation with SRL32 based controller's M1 / K also eliminates the use of input multiplexers. Hence, the logic functionality of MISTY1 and KASUMI odd / even rounds XOR operation is obtained.

Table 3 summarizes the initial values and the reconfigurable values of SRL32-1, 2, 3 and 4. The re-configuration delay of FL function and 32-bit XOR is 33 × clock cycles and 16 × clock cycles respectively. The additional delay of 1 × clock cycle for FL function is due to the reconfigurable values for which the data enable has to be de-asserted for one clock cycle. The RT of FL function thus dictates the delay for the proposed algorithm to dynamically modify the circuit operation from MISTY1 to KASUMI.

The FO function and the respective FI function for RLUT architecture are shown in Fig 5a and 5b respectively. The FO function performs round operations as well as EK generation for MISTY1 and KASUMI. The optimum operation of FO function is performed by reconfiguring the FI (S9 and S7 s-boxes) as well as enabling or disabling the $KO_i$ XORs by asserting the required secret key, $C_1$-$C_8$ constants and logic zero values. For MISTY1, the EKs are generated by nullifying the effect of $KO_i$ XORs using logic zeros to output the FI function. On the other

**Table 3. RLUT contents and RT for FL and 32-bit XOR.**

| Ftn | SRL32 Primitive | Initial Value | Reconfigurable Value | RT |
|---|---|---|---|---|
| FL | SRL32-1 | 6A6A6A6A | AA555555 | 33 × clock cycles |
| | SRL32-2 | A6A6A6A6 | AAAAAA55 | 33 × clock cycles |
| 32-bit XOR | SRL32-3 | 66660FF0 | C3C36666 | 16 × clock cycles |
| | SRL32-4 | 6666CC33 | C3C36666 | 16 × clock cycles |

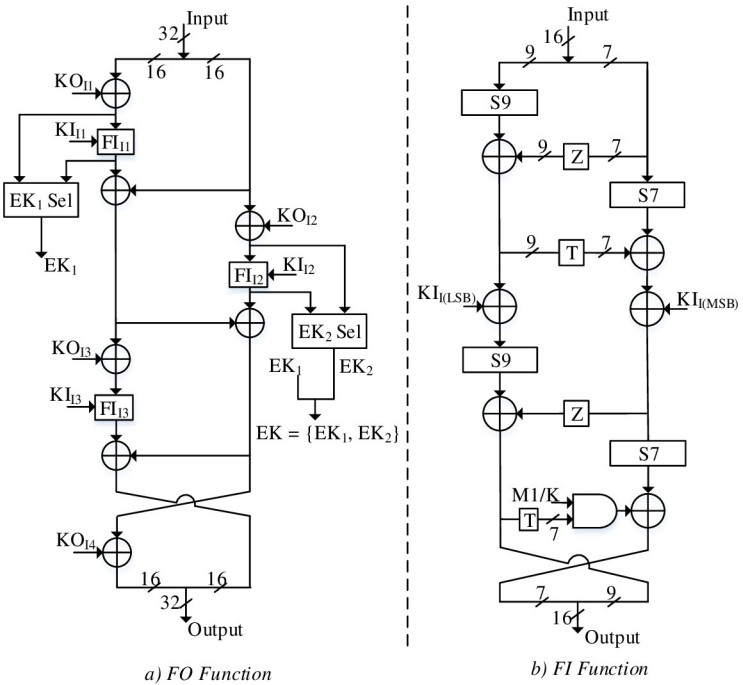

*a) FO Function*        *b) FI Function*

**Fig 5. a) FO Function, b) FI function.**

hand, the $KO_i$ XORs constituting $C_1$-$C_8$ constant values are utilized for EK generation of KASUMI. The output $EK_1$ and $EK_2$ for MISTY1 and KASUMI are thus saved in EK register.

The FI function consists of 4 × Feistel rounds with the last round consisting of S7 s-box as buffer SRLs for MISTY1. In addition, the logic XOR appended with S7 function is cancelled out by the output of AND gates driven through low M1 / K signal reproducing the 7-bit MSB output. Therefore, the FI function operates for MISTY1 by initiating MISTY1 S9 and S7 s-boxes for the first 3 Feistel rounds. Later, S9 and S7 s-boxes can be reconfigured for KASUMI by enabling the last Feistel round with M1 / K signal. The optimized S9 and S7 s-boxes are based on SRL32 primitive and are explained below.

## Implementation of reconfigurable S-boxes with RLUTs

The S7 s-box of (2nd round) FI function is designed using 5-input functions with output logic consisting of XOR for each S7 expression. The input and output bits relationship for MISTY1 and KASUMI S7 s-boxes is presented in Table 4. Based on input bits relationship for MISTY1 and KASUMI S7 s-box, a swapping circuit is required as depicted in Fig 6a. The output logic for KASUMI operation, on the other hand, requires reconfiguration as well as inversion for $y_3$, $y_4$, $y_5$ and $y_6$. In order to incorporate inversion, the output logic of S7 s-box is incorporated with additional M1 / K signal as shown in Fig 6b and therefore flips the respective bit for KASUMI operation.

The S7 s-box is designed with SRL32 consisting of initial contents for MISTY1. By inserting low M1 / K signal, the swapping and flipping of the input and output sides is discarded such that input and output circuits act as a buffer and thus S7 s-box operates for MISTY1. To upload the KASUMI S7 s-box contents, a cascaded reconfiguration data operation is performed. Thus, the contents of SRL32 for MISTY1 S7 s-box are re-utilized to configure 2 × S7

**Table 4. RLUT contents and reconfiguration time for FL and 32-bit XOR.**

| Input | | Output | |
|---|---|---|---|
| MISTY1 | KASUMI | MISTY1 | KASUMI |
| $x_0$ | $x_6$ | $y_0$ | $y_6'$ |
| $x_1$ | $x_5$ | $y_1$ | $y_2$ |
| $x_2$ | $x_4$ | $y_2$ | $y_5'$ |
| $x_3$ | $x_3$ | $y_3$ | $y_1$ |
| $x_4$ | $x_2$ | $y_4$ | $y_3'$ |
| $x_5$ | $x_1$ | $y_5$ | $y_0$ |
| $x_6$ | $x_0$ | $y_6$ | $y_4'$ |

s-boxes (KASUMI has 2 × S7 s-boxes in FI) in 32 clock cycles for KASUMI. This methodology represents a significant reduction in area for S7 s-boxes implementation.

The flow diagram for implementation of S9 s-boxes is shown in Fig 7. The S9 s-boxes are implemented by eliminating the Close Support Electronics (CSEs) and logically formulating the LUTs for MISTY1 and KASUMI algorithms. The LUTs are designed such that the output combination of MISTY1 / KASUMI S9 s-boxes drives the RLUTs with 4-bit input. Thus, SRL32 are employed at the output of each expression for S9 s-box to perform a MUX operation as well as XOR function for MISTY1 / KASUMI. The reconfiguration time of SRL32 for S9 s-box is 16 × clock cycles.

## Key scheduling

The key scheduling for RLUT architecture is presented in Fig 8. Since the round operations require 11 × 16-bit keys, the input multiplexers select the required secret keys and extended

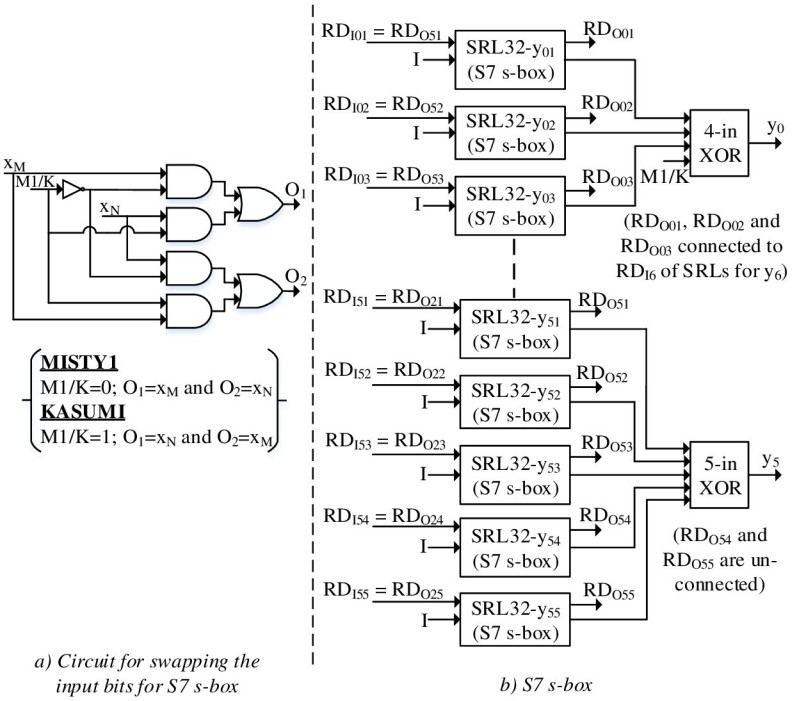

a) Circuit for swapping the input bits for S7 s-box

b) S7 s-box

**Fig 6. a) Circuit for swapping the inputs, b) S7 s-box.**

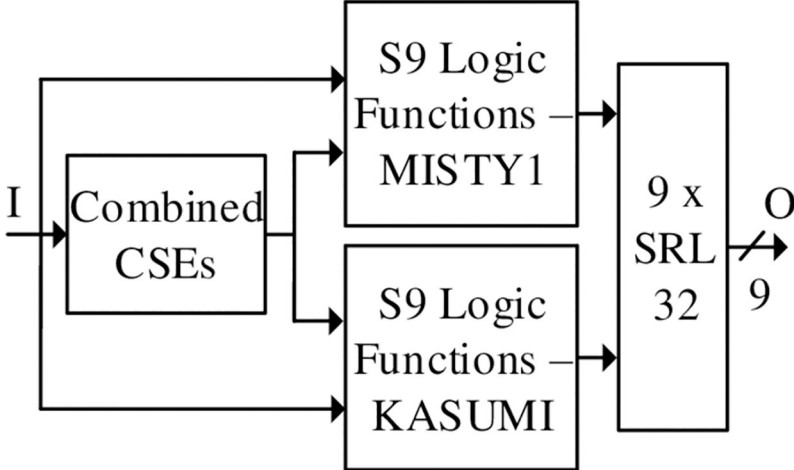

**Fig 7. S9 s-box.**

keys for respective rounds. However, the KASUMI secret keys differ from MISTY1 in terms of 1-bit, 5-bit, 8-bit and 13-bit circular left shift rotations.

The RLUTs are employed to act as buffers for MISTY1 and are reconfigured by Counter's $C_2$ and $C_3$ signals for KASUMI with RT of 16 clock cycles. The dynamic reconfigurable part also consists of $2 \times 16$-bit swapped keys for KASUMI FL Even function and is implemented

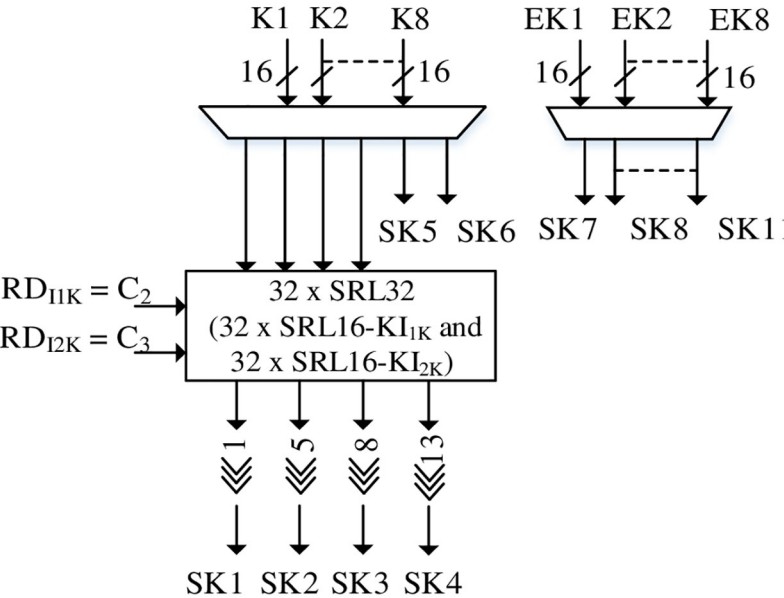

$$SK = \{SK1 \parallel SK2 \parallel SK3 \parallel \dots \parallel SK11\}$$

**Fig 8. Key scheduling for RLUT architecture.**

the same way as Fig 6a. Thus, the required 64-bit cipher text is generated for MISTY1 / KASUMI with the proposed key scheduling scheme.

## DPR based reconfigurable MISTY1 / KASUMI

A DPR based MISTY1 / KASUMI architecture is presented in Fig 9 and differs from RLUT hardware in terms of reconfiguration technique and EK generation / scheduling. The reconfiguration is carried out by dynamically uploading the partial bit stream stored in external hardware. For the proposed DPR architecture, the FPGA core is preset for MISTY1 algorithm (similar to RLUT architecture); however, the design can also be stipulated for vice versa implementation. This infers that the static logic only consists of common logic for MISTY1 and KASUMI and can be configured by uploading either of the partial bit streams for MISTY1 or KASUMI.

The primary advantage of this design is the power reduction compared to un-optimized equivalent parallel architectures for MISTY1 / KASUMI. The design is based on re-utilization of common logic and is configured as the static part whereas the partial bit streams consist of dynamic logic for the transformation functions to be uploaded on required MISTY1 /

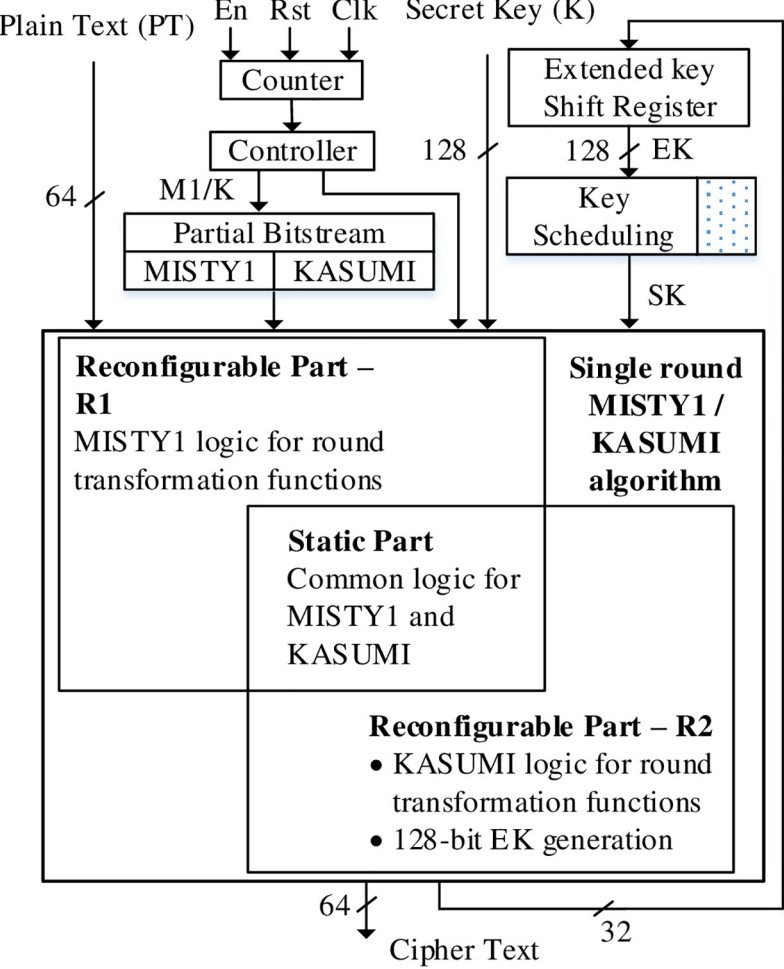

**Fig 9. DPR hardware architecture MISTY1 / KASUMI.**

KASUMI operations. The varying logic functionality therefore results in reconfigurable architecture.

## A single round MISTY1 / KASUMI algorithm

A single round MISTY1 / KASUMI algorithm for DPR architecture is presented in Fig 10a consisting of static and dynamic parts for 2 × FL functions and FO function (the dotted parts represent the dynamic logic). For FL functions, the logic combinations AND / OR / XORs are re-utilized; the reconfiguration can be carried out for KASUMI algorithm by implementing 2 × 1-bit circular left-shift. Similarly, 32-bit XOR can be implemented for single round MISTY1 / KASUMI by reconfiguring the input mixes.

Unlike RLUT architecture, a single round DPR algorithm employs FO (FI) function to generate EK (i.e., $EK_1$ and $EK_2$) only for MISTY1 algorithm whereas the EK for KASUMI is generated by independent i.e., reconfigurable key generation module. In addition, the FO function consists of static / dynamic logic parts for FI function as shown in Fig 10b. The reconfigurable FI function has S9 / S7 s-boxes and 7-bit XOR as reconfigurable parts and thus the 4th Feistel round is configured only for KASUMI FI function operation. The s-boxes S9 and S7 s-boxes are implemented using CSE methodology. For S7 s-box, the combinational logic is re-utilized by configuring the input / output bits. On the other hand, the S9 s-box consists of combined CSEs for MISTY1 / KASUMI as static logic whereas additional logic of S9 s-box can be dynamically configured for MISTY1 / KASUMI implementation. The S9 s-box logic details for static part and dynamic parts are shown in Table 5 showing 42% area reduction compared to non-optimized equivalent parallel S9 s-boxes implementation.

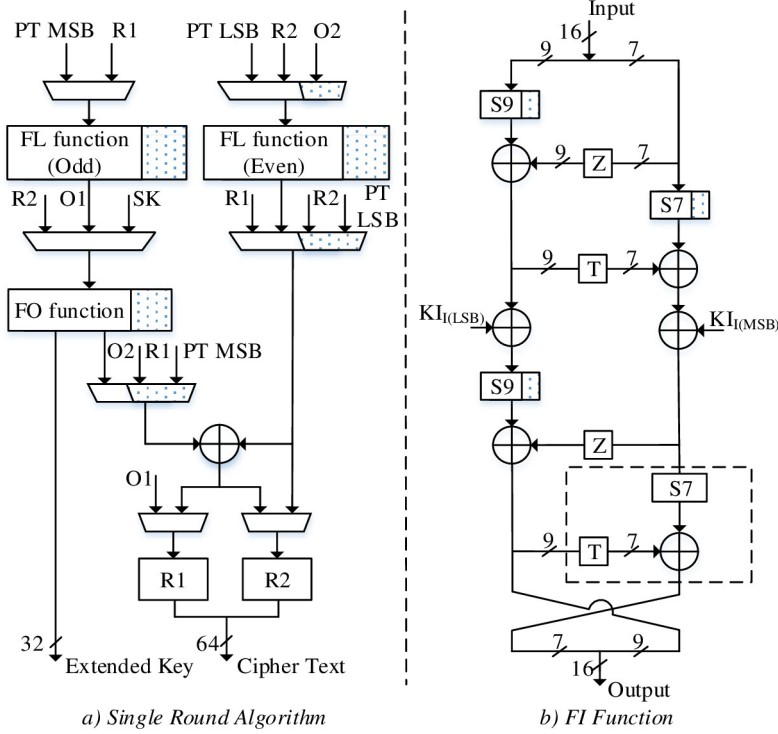

*a) Single Round Algorithm*          *b) FI Function*

**Fig 10. a) DPR based Single Round MISTY1 / KASUMI, b) FI function.**

**Table 5. % area reduction (NAND gates) for S9 s-box implementation.**

| Method | Static | Dynamic 1 | Dynamic 2 | Total | % Red |
|---|---|---|---|---|---|
| Prop. S9 | 152 | 148 | 138 | 438 | 42 |
| MISTY1 S9 | 386 | - | - | 758 | |
| KASUMI S9 | 372 | - | - | | |

## Extended key generation and key scheduling

The key scheduling scheme of the proposed DPR architecture is shown in Fig 11. It has been designed for dynamic operability such that the secret keys are applied through reconfigurable 1-bit, 5-bit, 8-bit, and 13-bit circular left shift operation whereas $2 \times 16$-bit keys of FL Even for MISTY1 are swapped in the dynamic region for KASUMI algorithm. The extended key of KASUMI is generated in 1 clock cycle by implementing $8 \times 16$-bit XORs in reconfigurable region for KASUMI algorithm. The 16-bit secret keys and extended keys are finally multiplexed to output the required $11 \times 16$-bit key for MISTY1 / KASUMI algorithm round operations.

## FPGA implementation—Results and analysis

The proposed reconfigurable hardware architectures are implemented on Xilinx Vertex 7 FPGA, XC7VX690T. Table 6 summarizes the analysis of the proposed designs based on the performance parameters including area utilization, power consumption, speed, and RT.

The RLUT architecture shows remarkable results with high throughput value and lesser RT of 364 Mbps and 445 nsec respectively compared to its counterpart DPR architecture with parametric values of 176 Mbps and 1.1 msec. Moreover, the occupied area and the power consumption of RLUT architecture is found to be 668 CLB slices and 460 mW respectively and are comparable to the proposed DPR architecture occupying 514 CLB slices with power dissipation of 354 mW. This signifies the advantages of RLUT architecture outperforming existing reconfigurable implementations.

The proposed DPR architecture, on the other hand, shows comparable performance with the DPR architectures of refs [27, 28, 30]. In terms of area, the proposed reconfigurable

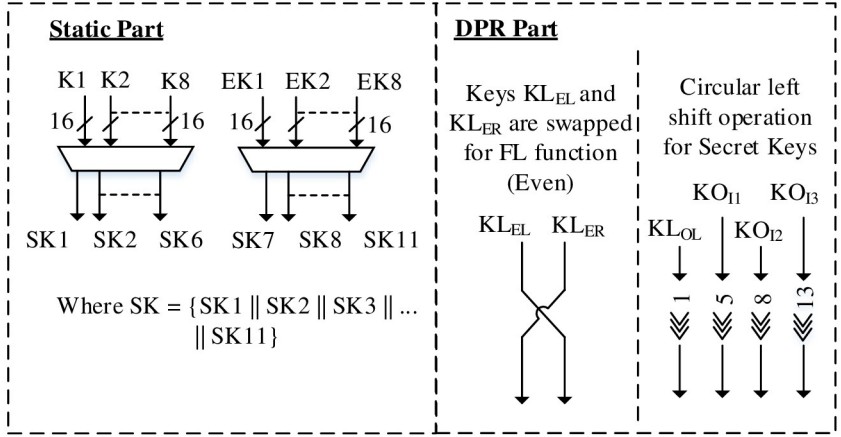

**Fig 11. Key scheduling.**

**Table 6. Performance analysis of hardware architectures (M = 10$^6$, G = 10$^9$, m = 10$^{-3}$, n = 10$^{-9}$) (*implementation for reference / comparison).**

| Ref. | Algorithm | Device/Tech | Area (Gates/CLB Slices) | Speed (bps) | Power | RT (sec) |
|---|---|---|---|---|---|---|
| [12] | MISTY1 | 180 nm | 3041 | 171 M | - | - |
| [12] | MISTY1 | 180 nm | 2331 | 166 M | - | - |
| [13] | MISTY1 | Virtex7 | 1265 | 16.3G | - | - |
| [14] | KASUMI | 180nm | 2640 | 122 M | - | - |
| [14] | KASUMI | Virtex7 | 1847 | 27.5 G | - | - |
| [16] | KASUMI | 110nm | 2990 | 110 M | - | - |
| [17] | MISTY1 / KASUMI | 180nm | 3481 | 130 M | - | - |
| [18] | SNOW 3G / ZUC | 65nm | 20,600 | 29.4 G | 17 mW | 1.11n |
| [27] | DPR Arch (AES/HELP) | Zynq | LUTs—3305 | - | 1.9 mW | 0.663m |
| | | | FFs—1601 | | | |
| | | | MUX—256 | | | |
| [28] | DPR Arch. (6 Algos) | Kintex7 | 1477 | - | 216 mW | - |
| [30] | DPR Arch (AES) | Zynq | 1279 | 26 G | - | - |
| Implementation for Reference / Comparison* | Single Round–MISTY1 & KASUMI | Virtex7 | 996 | 507 M | 688 mW | 13.5n |
| Proposed | RLUT Arch | Virtex7 | 668 | 364 M | 460 mW | 445n |
| | DPR Arch | Virtex7 | 514 | 176 M | 354 mW | 1.1m |

architectures have lesser CLB slices whereas reconfiguration time of our RLUT architecture is lesser as compared to ref [27]. It is worth mentioning that no DPR architectures have been proposed for MISTY1/KASUMI; the studies have been found only for implementation of AES with HELP algorithm. Contrary to DPR architecture, the novel RLUT architecture has never been proposed for implementation of two or more cryptographic circuits. Therefore, in terms of run-time reconfigurability, the RLUT architecture is novel and has been compared with DPR / stand-alone hardware architectures.

Finally, to obtain a fair comparison, a single round parallel MISTY1 and KASUMI architecture is also implemented for reference. The RLUT / DPR schemes are both area and power efficient compared to single round parallel MISTY1 / KASUMI whereas higher reconfiguration time of our architectures can be countered with the key advantage of dynamic reconfiguration. Thus, the proposed architectures outperform the straight-forward implementation and have core applications in sensor networks and military wireless gadgets.

## Conclusion

This paper presents state-of-the-art dynamically reconfigurable hardware architectures for MISTY1 and KASUMI block ciphers. The DPR and SRL based FPGA schemes are explored for run-time reconfiguration of MISTY1 and KASUMI algorithms by utilizing common logic. The RLUT architecture signifies core advantages of dynamic reconfiguration, high throughput, and lesser reconfiguration time with comparable parametric values of area / power of the proposed DPR architecture. The hardware designs are highly suitable for military applications and wireless sensor networks. The proposed methodologies can be extended to security architecture for UMTS networks employing multiple algorithms for confidentiality and integrity in 3g and 4g systems. To sum up, this work can be regarded as a significant development in circuit design and cryptography leading to future generation security architecture designs.

## Author Contributions

**Conceptualization:** Huang Jiexian.

**Data curation:** Yasir Khizar.

**Formal analysis:** Yasir Khizar.

**Investigation:** Yasir Khizar.

**Methodology:** Yasir Khizar.

**Project administration:** Zain Anwar Ali.

**Resources:** Raza Hasan, Muhammad Salman Pathan.

**Software:** Huang Jiexian, Raza Hasan, Muhammad Salman Pathan.

**Supervision:** Zain Anwar Ali.

**Writing – original draft:** Raza Hasan, Muhammad Salman Pathan.

**Writing – review & editing:** Muhammad Salman Pathan.

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
