## [Decision Letter · Decision Letter 0]

13 Apr 2023

PONE-D-22-34109On the Dynamic Reconfigurable Implementations of MISTY1 and KASUMI Block CiphersPLOS ONE

Dear Dr. ALI,

Thank you for submitting your manuscript to PLOS ONE. After careful consideration, we feel that it has merit but does not fully meet PLOS ONE’s publication criteria as it currently stands. Therefore, we invite you to submit a revised version of the manuscript that addresses the points raised during the review process.

We look forward to receiving your revised manuscript.

Kind regards,

Je Sen Teh

Academic Editor

PLOS ONE

Journal Requirements:

Muhammad Salman Pathan 

None 

5. Please note that in order to use the direct billing option the corresponding author must be affiliated with the chosen institute. Please either amend your manuscript to change the affiliation or corresponding author, or email us at plosone@plos.org with a request to remove this option.

Additional Editor Comments:

We would first like to apologize to the authors for the delay in processing the manuscript. The reviewer has raised some queries related to the implementation details of the block ciphers and performance measures that I would like the authors to address.

Reviewers' comments:

Reviewer's Responses to Questions

**Comments to the Author**

1. Is the manuscript technically sound, and do the data support the conclusions?

Reviewer #1: Yes

2. Has the statistical analysis been performed appropriately and rigorously? 

Reviewer #1: Yes

3. Have the authors made all data underlying the findings in their manuscript fully available?

Reviewer #1: No

4. Is the manuscript presented in an intelligible fashion and written in standard English?

Reviewer #1: Yes

5. Review Comments to the Author

Reviewer #1: The work publishes DPR implementations of MISTY1 and KASUMI block ciphers. This is an improvement over their previous works which were static FPGA implementations of either of the ciphers. Overall, the work adds some important incremental value to the state-of-the-art. My comments:

Do the MISTY1 and KASUMI implementations use some of the built-in IP (DSP blocks, for instance) to improve performance? If not, would it be feasible for these implementations?

Are there technology agnostic measures that can be used to evaluate the works in Table 6?

1.9 mJ is Energy (under the Power column)

I would like the DPR hardware efficiency vs. delay requirements results as is the norm in DPR literature. Another important result is the Area vs. delay requirements measurements.

Fig 6 and 7 can appear side-by-side?

The abbreviation PUF and CSE have been used before expansion.

Grammar :

Consider rewriting with improved language / grammar. Break the long sentence into multiple parts, perhaps  Section 1. Para 2. Sentence 1.

Section 1. Para 3. Use one of the terms -- features / design constraints. drawbacks / shortcomings.

Long sentences that can be broken into smaller segments:

- Section 1. Para 3. The two algorithms ...

- Section 1. Para 4. In this regard, ...

6. PLOS authors have the option to publish the peer review history of their article (what does this mean?). If published, this will include your full peer review and any attached files.

Reviewer #1: No

---

## [Author Response · Author response to Decision Letter 0]

22 Jun 2023

Do the MISTY1 and KASUMI implementations use some of the built-in IP (DSP blocks, for instance) to improve performance? If not, would it be feasible for these implementations?

No, DSP blocks is not feasible in this scenario because we need parallel processing in implementations of MISTY1 and KASUMI Block Ciphers

Are there technology agnostic measures that can be used to evaluate the works in Table 6? 1.9 mJ is Energy (under the Power column)

Yes, we have compared the performance measures of the different Hardware Architectures available. Also 1.9mJ is corrected as 1.9mW

I would like the DPR hardware efficiency vs. delay requirements results as is the norm in DPR literature. Another important result is the Area vs. delay requirements measurements.

Fig 6 and 7 can appear side-by-side?

Corrected 

The abbreviation PUF and CSE have been used before expansion.

Corrected

Grammar:

Consider rewriting with improved language / grammar. Break the long sentence into multiple parts, perhaps  Section 1. Para 2. Sentence 1.

Section 1. Para 3. Use one of the terms -- features / design constraints. drawbacks / shortcomings.

Long sentences that can be broken into smaller segments:

- Section 1. Para 3. The two algorithms ...

- Section 1. Para 4. In this regard, ...

---

## [Decision Letter · Decision Letter 1]

10 Aug 2023

PONE-D-22-34109R1On the Dynamic Reconfigurable Implementations of MISTY1 and KASUMI Block CiphersPLOS ONE

Dear Dr. ALI,

Thank you for submitting your manuscript to PLOS ONE. After careful consideration, we feel that it has merit but does not fully meet PLOS ONE’s publication criteria as it currently stands. Therefore, we invite you to submit a revised version of the manuscript that addresses the points raised during the review process.

 As of this moment, the manuscript is considered provisionally accepted pending the minor revisions listed below.

We look forward to receiving your revised manuscript.

Kind regards,

Je Sen Teh

Academic Editor

PLOS ONE

Journal Requirements:

Additional Editor Comments:

The first reviewer is satisfied with the corrections made while the second reviewer has proposed some minor improvements including language correction. I suggest that the manuscript be accepted pending minor corrections based on Rev2 comments.

Reviewers' comments:

Reviewer's Responses to Questions

**Comments to the Author**

1. If the authors have adequately addressed your comments raised in a previous round of review and you feel that this manuscript is now acceptable for publication, you may indicate that here to bypass the “Comments to the Author” section, enter your conflict of interest statement in the “Confidential to Editor” section, and submit your "Accept" recommendation.

Reviewer #1: All comments have been addressed

Reviewer #2: All comments have been addressed

2. Is the manuscript technically sound, and do the data support the conclusions?

Reviewer #1: Yes

Reviewer #2: Partly

3. Has the statistical analysis been performed appropriately and rigorously? 

Reviewer #1: N/A

Reviewer #2: No

4. Have the authors made all data underlying the findings in their manuscript fully available?

Reviewer #1: Yes

Reviewer #2: Yes

5. Is the manuscript presented in an intelligible fashion and written in standard English?

Reviewer #1: Yes

Reviewer #2: No

6. Review Comments to the Author

Reviewer #1: All of my review comments have been addressed. My recommendation for this manuscript is an Accept .

Reviewer #2: 1. Result should be elaborated more in context of abstract

2. Novelty should be highlighted with proper showcase of results

3. Authors should elaborate or highlight the gap between the existing design and the proposed design.

4. Justify the performance metrics used and the result comparison. More comparative analysis need to be provided.

5. Grammar check need to be done throughout the text

7. PLOS authors have the option to publish the peer review history of their article (what does this mean?). If published, this will include your full peer review and any attached files.

Reviewer #1: No

Reviewer #2: No

---

## [Editor Report · Decision Letter 2]

30 Aug 2023

On the Dynamic Reconfigurable Implementations of MISTY1 and KASUMI Block Ciphers

PONE-D-22-34109R2

Dear Dr. ALI,

We’re pleased to inform you that your manuscript has been judged scientifically suitable for publication and will be formally accepted for publication once it meets all outstanding technical requirements.

Kind regards,

Je Sen Teh

Academic Editor

PLOS ONE

Additional Editor Comments (optional):

All concerns have been addressed by the authors.
---

## [Editor Report · Acceptance letter]

21 Sep 2023

PONE-D-22-34109R2 

On the Dynamic Reconfigurable Implementations of MISTY1 and KASUMI Block Ciphers 

Dear Dr. Ali:

I'm pleased to inform you that your manuscript has been deemed suitable for publication in PLOS ONE. Congratulations! Your manuscript is now with our production department. 

Kind regards, 

on behalf of

Dr. Je Sen Teh 

Academic Editor

PLOS ONE